# Ocean iron cycle feedbacks decouple atmospheric CO₂ from meridional overturning circulation changes

Jonathan Maitland Lauderdale [1] ✉

The ocean's Meridional Overturning Circulation (MOC) brings carbon- and nutrient-rich deep waters to the surface around Antarctica. Limited by light and dissolved iron, photosynthetic microbes incompletely consume these nutrients, the extent of which governs the escape of inorganic carbon into the atmosphere. Changes in MOC upwelling may have regulated Southern Ocean outgassing, resulting in glacial-interglacial atmospheric CO₂ oscillations. However, numerical models that explore this positive relationship do not typically include a feedback between biological activity and abundance of organic chelating ligands that control dissolved iron availability. Here, I show that incorporating a dynamic ligand parameterization inverts the modelled MOC-atmospheric CO₂ relationship: reduced MOC nutrient upwelling decreases biological activity, resulting in scant ligand production, enhanced iron limitation, incomplete nutrient usage, and ocean carbon outgassing, and vice versa. This first-order response suggests iron cycle feedbacks may be a critical driver of the ocean's response to climate changes, independent of external iron supply.

The role of ocean circulation in the global carbon cycle and climate is a complex interplay of physical, biogeochemical, and ecological processes[1,2]. To grow, phytoplankton require sunlight, inorganic carbon, large relative quantities of macronutrients, such as nitrate and phosphate, as well as much smaller amounts of micronutrients, such as iron. A fraction of the organic matter produced in the surface ocean forms sinking particles ("marine snow") that are gradually broken down with depth by bacteria in the intermediate- and deep-ocean, releasing their constituents back into the seawater[3]. These resources are returned to the surface over long timescales as part of the global Meridional Overturning Circulation (MOC), the network of deep-ocean and surface currents that traces water masses on a continuous circuit of the global ocean[4]. The MOC can be conceptualized as two connected cells: (a) the Atlantic Meridional Overturning Circulation (AMOC), which is driven by sinking of dense waters in the Nordic Seas and North Atlantic, and (b) the Southern Ocean Meridional Overturning Circulation (SMOC), which is comprised of dense bottom water sinking adjacent to Antarctica. Dense watermass transformation and upwelling in the Indian and Pacific Oceans due to turbulent mixing begins the journey back towards the upper ocean[4]. These oldest waters carry the highest concentrations of remineralized carbon and nutrients. Both cells are ulimately connected back to the surface, and the atmosphere, by Southern Ocean upwelling driven by strong westerly winds, air-sea heat fluxes, and freshwater exchanges. The strength of the Atlantic and Southern Ocean MOC cells are similar in magnitude, approximately 20 Sv[5].

Unlike lower latitudes, where macronutrients and inorganic carbon are generally rapidly depleted by biological activity near the surface, high concentrations of macronutrients and inorganic carbon are sustained in the Southern, Equatorial and Northern Pacific Oceans as a consequence of local biological limitation by lack of other resources such as light and/or the micronutrient iron[6]. Iron is not very soluble in seawater[7] and may be rapidly lost from solution by precipitation or by scavenging, that is adsorption onto sinking particles[8]. Therefore older

[1]Department of Earth, Atmospheric and Planetary Sciences, Massachusetts Institute of Technology, 77 Massachusetts Avenue, Cambridge 02139 MA, USA.
✉e-mail: jml1@mit.edu

upwelled waters are typically low in iron, having been exposed to prolonged iron depletion processes. In addition, the remote Southern Ocean receives little external iron input from atmospheric dust or continental sources[9]. Thus, upwelled macronutrients in the Southern Ocean, Equatorial, and North Pacific, are incompletely consumed, while the accompanying upwelled inorganic carbon may escape from the ocean, promoting higher atmospheric carbon dioxide ($CO_2$) levels.

Changing the strength of the MOC, and the Southern Ocean upwelling that connects the deep ocean inorganic carbon store to the atmosphere has long been identified as a key driver of past climatic changes such as the glacial-interglacial cycles[10–15]: slowing upwelling and reducing delivery of inorganic carbon and macronutrients to the Southern Ocean surface is thought to enable more complete resource consumption by microbes limited by iron and light, therefore lowering the evasion of upwelled $CO_2$ to the atmosphere. Increased Antarctic aeolian dust deposition[16] could have further enhanced Southern Ocean nutrient consumption during glacial periods[17]. Conversely, increasing upwelling enhances the Southern Ocean surface supply of inorganic carbon and macronutrients from the abyss, reduces the extent of Southern Ocean resource consumption by iron- and light-limited microbes, and promotes $CO_2$ outgassing. Early modelling studies highlighting the role of the Southern Ocean and its effect on atmospheric $CO_2$ [e.g., refs. 10–13] did not explicitly include iron cycling in their idealized box models. Nevertheless, more recent simulations that do include iron cycling still arrive at essentially the same result [e.g., refs. 1,2,18]: change in atmospheric $CO_2$ is proportional to change in MOC strength.

Over 99% of oceanic iron is associated with metal-complexing organic molecules, collectively known as ligands[19–23], that protect dissolved iron from loss by scavenging[8]. The diverse, incompletely characterised, ligand pool consists of siderophores, exuded polysaccharides, porphyrins, degraded protein remnants, and humic substances[24]. Ligands are produced by the growth and decay of marine microbes, which in turn depends on iron concentrations, establishing a reinforcing feedback loop[22] in which the creation of ligands by microbial activity stabilizes a greater quantity of iron, which may accumulate rather than be lost by scavenging. More available iron enables further uptake of nutrients for growth, additional ligand production, and enhanced iron accumulation. The feedback is arrested when another resource, such as light or macronutrients, becomes limiting. This coupling between biology and trace metal availability has not been considered in numerical model studies of changing MOC and Southern Ocean upwelling on ocean-atmosphere carbon partitioning.

How does atmospheric $CO_2$ respond to changes in Southern Ocean upwelling and coupled feedbacks within the ocean carbon, macronutrient, and iron cycles? I present a suite of idealized box model simulations of ocean circulation, trace metal biogeochemistry, and carbon cycling [refs. 25,26, Methods] in the spirit of the influential idealized models employed previously[10–13]. Within this framework, I can quantify the effect of dynamic ligand concentrations on atmospheric $CO_2$ as a result of increasing or decreasing MOC strength.

## Results

### Sensitivity of atmospheric $pCO_2$ to MOC strength

The first model ensemble has no feedback between biological activity and trace metal cycling, with ligand concentrations fixed at a uniform global concentration representative of the modern ocean [2 nmol kg$^{-1}$,27]. This results in the recognisable positive relationship between Southern Ocean upwelling and atmospheric $CO_2$ across a broad range of overturning rates (Fig. 1a) from a weak MOC with 8.0 Sv of Southern Ocean upwelling to a vigorous double-strength circulation of 40.0 Sv Southern Ocean upwelling. The increase in atmospheric $pCO_2$ amounts to 19 $\mu$atm across a realistic range of AMOC transports at the last glacial maximum [8.0–25.0 Sv, e.g., refs. 28–32. At Southern Ocean upwelling less than 8.0 Sv, approaching MOC collapse, the system enters a different regime that will not be considered here, where slow transports of inorganic carbon and nutrients between the deep ocean and the surface create a highly isolated abyssal inorganic carbon store. At 8.0 Sv, the vast deep Pacific Ocean is ventilated by a net transport of only 1.6 Sv (Methods), creating a residence time on the order of 10 kyrs that effectively traps inorganic carbon and nutrients, leading to decline in atmospheric $pCO_2$ of ~50–60 $\mu$atm, irrespective of surface macro- or micronutrient limitation. In the second model ensemble, the same experiments are repeated, this time enabling the feedback between biological activity and the iron cycle via emergent organic ligand concentrations (Methods[22]). The net result is elevated atmospheric $pCO_2$ at low MOC and lower atmospheric $pCO_2$ with strong MOC (Fig. 1b)−the opposite relationship to the ensemble with fixed ligand concentration equating to a 52 $\mu$atm decrease in atmospheric $pCO_2$ across a realistic range of AMOC transports at the last glacial maximum [8.0–25.0 Sv, e.g., refs. 28–32]. The inverted MOC-atmospheric $pCO_2$ relationship with dynamic ligand concentrations

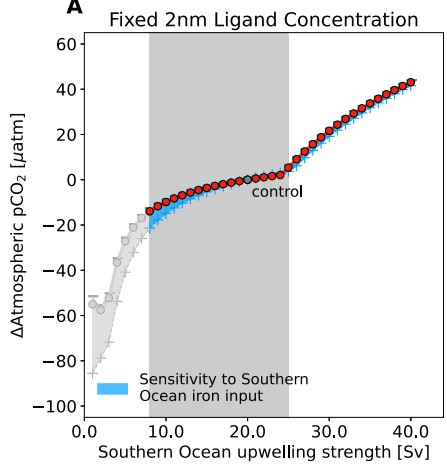
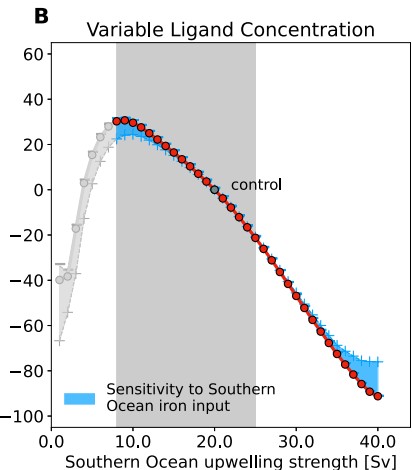

**A** Fixed 2nm Ligand Concentration

**B** Variable Ligand Concentration

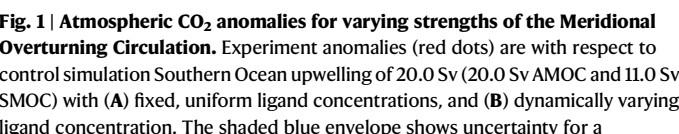

**Fig. 1 | Atmospheric CO₂ anomalies for varying strengths of the Meridional Overturning Circulation.** Experiment anomalies (red dots) are with respect to control simulation Southern Ocean upwelling of 20.0 Sv (20.0 Sv AMOC and 11.0 Sv SMOC) with (**A**) fixed, uniform ligand concentrations, and (**B**) dynamically varying ligand concentration. The shaded blue envelope shows uncertainty for a 20 × increase (plus symbol) or decrease (minus symbol) in surface iron input to the Southern Hemisphere, while the grey box highlights the range of realistic AMOC transports for the last glacial maximum [8.0–25.0 Sv, e.g., refs. 28–32]. Source data are provided as a Source Data file.

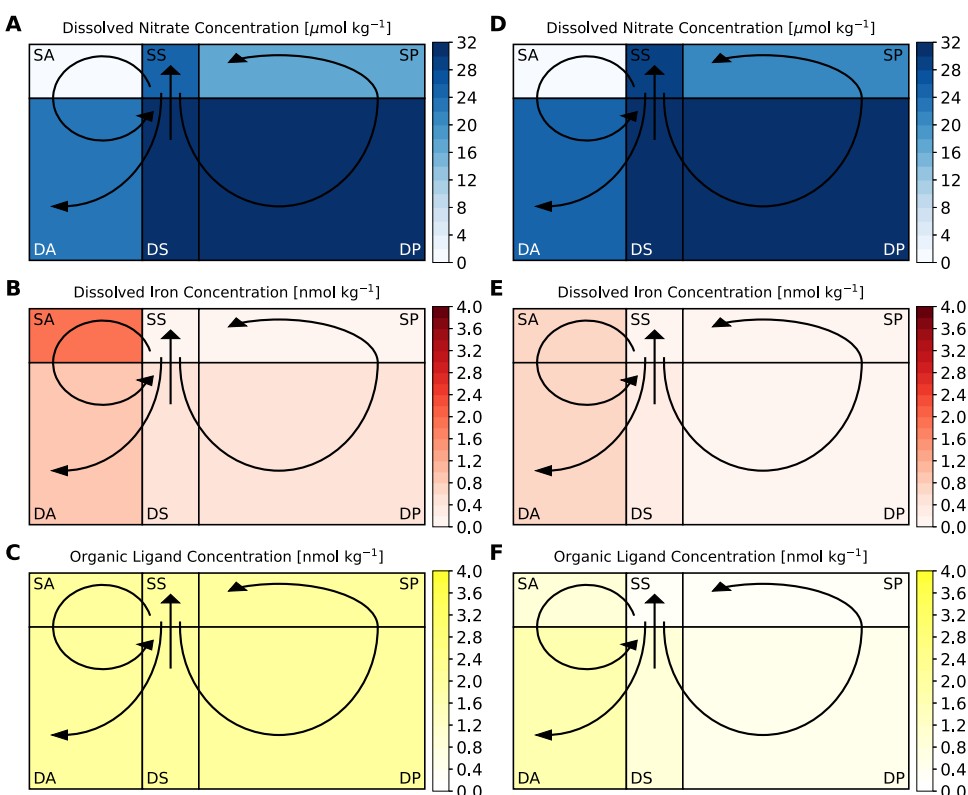

**Fig. 2 | Steady-state results for model simulations with weak Southern Ocean upwelling strength (8.0 Sv).** Ligand levels are represented as fixed, uniform concentrations (left column), or are allowed to dynamically vary (right column). Concentrations are (**A, D**) nitrate ($\mu$mol N kg$^{-1}$), (**B, E**) iron (nmol Fe kg$^{-1}$), and (**C, F**) organic ligands (nmol L kg$^{-1}$). Black arrows indicate the overturning circulation between boxes representing the Surface Atlantic (SA), Surface Southern (SS), Surface Pacific (SP), Deep Atlantic (DA), Deep Southern (DS), and Deep Pacific (DP) Oceans. Boxes are not drawn to scale in the vertical. Source data are provided as a Source Data file.

is robust to large changes in Southern Hemisphere aeolian iron deposition[17], illustrated by the shaded region in Fig. 1.

## Oceanic response to MOC changes with fixed and dynamic ligands

A sluggish MOC with low Southern Ocean upwelling is most commonly associated with cool glacial climates[14,15]. Under current thinking, with fixed ligand concentration and no feedback, reducing the strength of the MOC (Fig. 1A) from the 20.0 Sv control to a weak 8.0 Sv decreases the upwelling supply of macronutrients and iron to the surface Southern Ocean (Fig. 2, and Supplementary Figs. S1, 2). Macronutrient abundance in the Southern Ocean (Fig. 2A) remains roughly the same, 24.0 $\mu$mol N kg$^{-1}$ compared to 22.6 $\mu$mol N kg$^{-1}$ in the control (Methods), even though biological activity declines by nearly 69%, indicating significantly more efficient nutrient consumption. In the Pacific there is a similar change, with 39% reduced biological production coupled to lower AABW upwelling, resulting in decreased macronutrient concentrations (17.4 $\mu$mol N kg$^{-1}$ compared to 21.6 $\mu$mol N kg$^{-1}$ in the control). Global nutrient use efficiency (see Methods) increases 13%, preventing more upwelled inorganic carbon from outgassing, lowering atmospheric CO$_2$ levels from 280 $\mu$atm to 266 $\mu$atm. The Surface Southern Ocean and Pacific Oceans remain iron-limited as micronutrient concentrations are drawn down to low levels (Fig. 2B), inhibiting the further uptake of remaining inorganic carbon. Lower MOC strength also reduces the upper-ocean delivery of nitrate from the Southern and Pacific Oceans to the macronutrient-limited Atlantic Ocean. Nitrate concentrations are still fully drawn down in the Atlantic Ocean, but the 65% lower biological activity due to heightened macronutrient limitation means less iron consumption, increasing the local standing stock by 1.54 nmol Fe kg$^{-1}$ (nearly 7 ×). Excess iron is exported to the Deep Atlantic, which also receives hydrothermal iron input but, although provided protection by the 2 nmol L kg$^{-1}$ constant ligand concentration, is subsequently partially lost by scavenging due to slower transit in the AMOC resulting in overall average deep ocean iron concentrations declining from 0.71 nmol Fe kg$^{-1}$ to 0.63 nmol Fe kg$^{-1}$. Lower deep ocean iron concentration and slower MOC reduce the Southern Ocean upwelling iron supply, enhancing surface iron limitation and thus constraining the extent of macronutrient and inorganic carbon consumption in the Southern and Pacific Oceans, limiting atmospheric CO$_2$ reduction.

Alternatively, when these experiments are repeated with dynamic ligand concentrations, Southern Ocean macronutrient levels increase to 28.7 $\mu$mol N kg$^{-1}$ (Fig. 2D, and Supplementary Figs. S3, 4) from 17.5 $\mu$mol N kg$^{-1}$, and the Southern Ocean is ultimately a source of CO$_2$ to the atmosphere. This is because reduced nutrient upwelling causes a 92% reduction in biological activity, resulting in a decline in organic ligand concentration from 3.33 nmol L kg$^{-1}$ to 0.34 nmol L kg$^{-1}$ (Fig. 2F). Similarly, in the Pacific, a 58% decline in biological activity due to reduced nutrient supply results in a decline in ligand concentration from 0.72 nmol L kg$^{-1}$ to 0.23 nmol L kg$^{-1}$. Without sufficient protection by complexation with ligands, scarce iron is significantly scavenged and lost (Fig. 2E), resulting in the Southern and Pacific Ocean surface boxes experiencing heightened iron limitation and accumulation of unused macronutrients. There is a 15% overall decline in the extent of nutrient usage, which leads to greater inorganic carbon outgassing from the ocean, driving atmospheric CO$_2$ levels upwards from

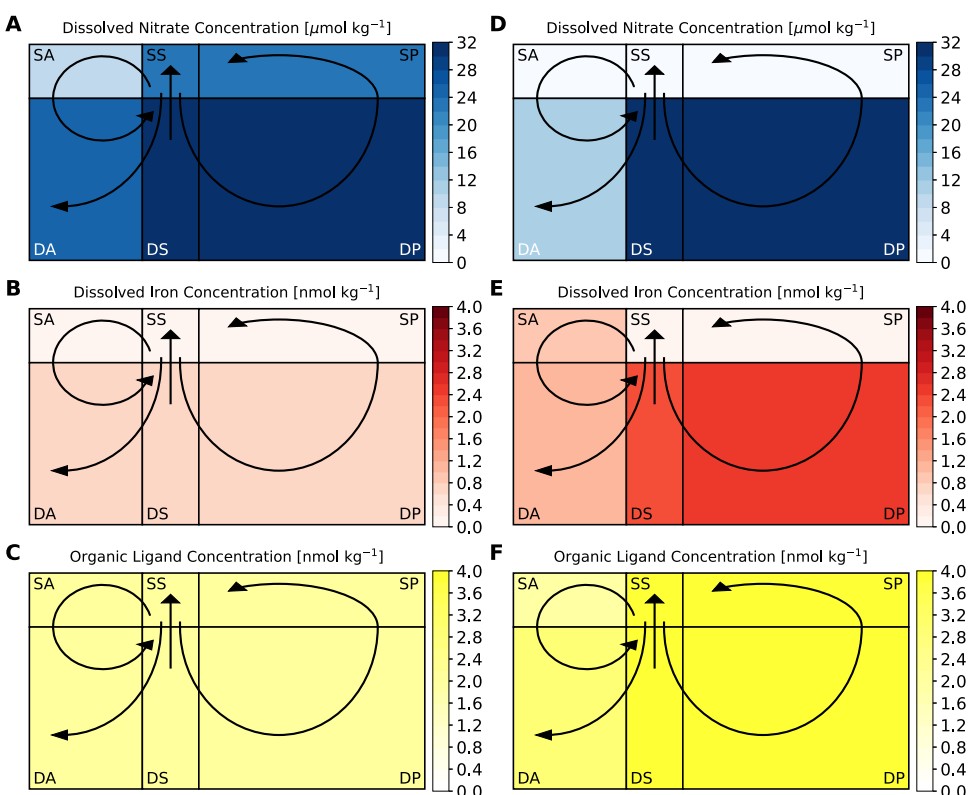

**Fig. 3 | Steady-state results for model simulations with strong Southern Ocean upwelling strength (40.0 Sv).** Ligand levels are represented as fixed, uniform concentrations (left column), or are allowed to dynamically vary (right column). Concentrations are (**A**, **D**) nitrate ($\mu$mol N kg$^{-1}$), (**B**, **E**) iron (nmol Fe kg$^{-1}$), and (**C**, **F**) organic ligands (nmol L kg$^{-1}$). Black arrows indicate the overturning circulation between boxes representing the Surface Atlantic (SA), Surface Southern (SS), Surface Pacific (SP), Deep Atlantic (DA), Deep Southern (DS), and Deep Pacific (DP) Oceans. Boxes are not drawn to scale in the vertical. Source data are provided as a Source Data file.

280 $\mu$atm to 310 $\mu$atm. Biological activity in the Atlantic Ocean is reduced by 53% and becomes co-limited by iron as well as nitrate, due to low ligand and, therefore, iron concentrations, as well as net reduced macronutrient supply from the Southern and Pacific Oceans (concentration increases, but MOC transport decreases). Finally, with very low ligand concentration, hydrothermal iron input to the deep ocean is heavily scavenged, resulting in deep ocean iron concentration falling from 1.00 nmol Fe kg$^{-1}$ to 0.33 nmol Fe kg$^{-1}$. Thus, there is much less iron supply in Southern Ocean upwelling, reinforcing inefficient nutrient consumption and promoting further evasion of $CO_2$ from the ocean to the atmosphere.

The sign of atmosphere-ocean carbon partitioning, under fixed and dynamic ligand scenarios, is reversed when the strength of the MOC and Southern Ocean upwelling is doubled from 20.0 Sv in the control to 40.0 Sv (Fig. 3).

For the model with prescribed ligand concentrations, increasing the rate of upwelling of deep ocean inorganic carbon, macronutrients, and iron to the Surface Southern Ocean increases biological activity by 2×as a consequence of macronutrient delivery and iron supply from deep-ocean hydrothermal vents, supported by fixed ligand concentrations in excess of the average - 0.70 nmol Fe kg$^{-1}$ in the deep ocean. However, the Southern and Pacific Oceans remain iron-limited, so surface macronutrient concentrations remain high (22.3 $\mu$mol N kg$^{-1}$ and 22.6 $\mu$mol N kg$^{-1}$, respectively, compared to 22.6 $\mu$mol N kg$^{-1}$ and 21.6 $\mu$mol N kg$^{-1}$ in the control, Fig. 3A), causing the Southern Ocean to remain a source of $CO_2$ to the atmosphere. Unused Southern Ocean and Pacific macronutrients exported into the Atlantic Ocean are partially consumed, causing a 32% increase in biological production.

However, surface iron concentrations become exhausted (declining from 0.27 nmol Fe kg$^{-1}$ to 0.00 nmol Fe kg$^{-1}$, Fig. 3B) before complete resource consumption, which allows accumulation of surface North Atlantic nitrate and inorganic carbon. The Atlantic Ocean transition from macronutrient limitation to iron limitation triggers acceleration of atmospheric $CO_2$ increase (Fig. 1A). Inefficient global nutrient and inorganic carbon consumption (a decline in completeness of macronutrient use of 17%) leads to greater ocean carbon outgassing, increasing atmospheric $pCO_2$ levels from 280 $\mu$atm to 323 $\mu$atm.

When ligand concentrations are dynamic, on the other hand, the Southern Ocean becomes a sink of atmospheric $CO_2$. Productivity in the Southern and Pacific Oceans is enhanced 4 and 5×, respectively, due to the increased upwelling of nutrients and iron from the deep ocean, which leads to the generation and transport of additional organic ligands, increasing global average concentrations to over 6.0 nmol L kg$^{-1}$ (Fig. 3F). Substantially greater protection from loss by scavenging increases the standing stock of bioavailable iron, especially in the deep ocean, from 1.00 nmol Fe kg$^{-1}$ to 1.97 nmol Fe kg$^{-1}$ (Fig. 3E). On upwelling into the Southern and Pacific Oceans, the enhanced supply of chelated iron from the deep ocean more than compensates for the greater delivery of macronutrients to the surface, resulting in a significant drawdown of both resources (Fig. 3D, E). Nitrate concentration in the surface Southern Ocean decreases from 17.50 $\mu$mol N kg$^{-1}$ in the control to 1.86 $\mu$mol N kg$^{-1}$, while concentrations in the Pacific decline from 19.5 $\mu$mol N kg$^{-1}$ in the control to 0.15 $\mu$mol N kg$^{-1}$, increasing the extent of consumption globally to - 100%, which prevents upwelled inorganic carbon from outgassing and results in lowering atmospheric $CO_2$ from 280 $\mu$atm to 189 $\mu$atm, close to the lower

bound of atmospheric $pCO_2$ due to bioligical activity calculated in ref. 11. Nitrate transport to the North Atlantic significantly declines, enhancing local macronutrient limitation and decreasing biological production by 61%. Excess iron from the Atlantic Ocean is transported into the Deep Atlantic, where it accumulates with additional iron from hydrothermal sources protected by elevated ligand concentrations. This eventually upwells in the Southern Ocean, reinforcing complete nutrient and inorganic carbon consumption.

Including the feedback between biological activity, organic chelating ligand abundance, and iron availability decouples atmospheric $pCO_2$ anomalies from MOC strength and Southern Ocean upwelling changes in this model. Instead of the positive functional relationship established in models that omit the cycling of iron or that include iron cycling with fixed, uniform ligand concentrations, including dynamic ligand concentrations produces a negative relationship between MOC strength and atmospheric $CO_2$ levels as a result of interaction between biological activity driven by nutrient delivery rates in the Southern and Pacific Oceans, emergent ligand concentrations, and the bioavailable standing stock of dissolved iron.

### Robustness of the MOC–atmospheric $CO_2$ relationship

Changes in external aeolian dust deposition to the Southern Ocean are hypothesized to significantly increase iron supply during glacial periods, stimulating iron-limited biological production in the Southern Ocean to consume unused nutrients and trap $CO_2$ in the ocean more efficiently[17]. Conversely, more humid conditions projected for the future climate reduce the aeolian iron flux[9] and could lead to enhanced ocean $CO_2$ outgassing[33]. The experiments with fixed and dynamic ligands were repeated with Southern Hemisphere surface iron flux increased or decreased by $20 \times$[9]. At low MOC strength with both fixed and dynamic ligands, elevated Southern Hemisphere surface iron deposition partially relieves Southern Ocean iron limitation (Fig. 2b, e) and enables further macronutrient consumption, resulting in an additional atmospheric $CO_2$ drawdown of between 10 and 20 $\mu$atm as hypothesized (Fig. 1, Supplementary Information and Figs. S1–4), but low light and low ligand abundance to support significantly higher iron concentrations prevent further $CO_2$ uptake. Overall, the dynamic ligand feedback is the primary factor affecting atmospheric $CO_2$ levels, with the "iron hypothesis"[17] playing a secondary, additional role in sluggish MOC scenarios.

Paleoclimate observations and simulations provide inconclusive evidence regarding the strength and depth-extent of the AMOC during the last glacial maximum [e.g., refs. 29,30,32,34–38]. The standard simulations have a fixed interface between the surface and deep Atlantic of 100 m, equivalent to the depth of the euphotic zone. The main result of opposing MOC-atmospheric $pCO_2$ correlations with fixed and dynamic ligand concentrations is robust to a significant change in the depth of the Surface Atlantic from 100 m to 2000 m (Supplementary Information, and Figs. S5–8), which affects Atlantic Ocean residence times. For fixed ligand concentrations, this results in higher iron accumulation and reduced ocean carbon outgassing, while with dynamic ligand concentration, this leads to increased ligand degradation, lower iron availability, and reduced ocean carbon uptake in the Southern and Pacific Oceans, particularly at strong Southern Ocean upwelling rates. The model uses prescribed transports[13,39], which implicitly include the effects of both advection by the MOC and fluxes due to turbulent mixing. The latter fluxes are relatively uncertain for the modern ocean and may have been more intense during glacial times due to enhanced tidal mixing [e.g., ref. 40]. An additional exchange of 3.0 Sv (~15% of control Southern Ocean upwelling strength) between adjacent boxes was applied (Supplementary Information, and Figs. S9–12), which does not significantly change the different MOC-atmospheric $pCO_2$ relationships, although at low Southern Ocean upwelling strengths, mixing is a significant continued source of nutrients and iron to the surface ocean, which results in

enhanced ocean carbon uptake with fixed ligand concentration and reduced ocean carbon outgassing with dynamic ligand concentrations.

The maximum rate of biological production associated with the availability of macro- and micronutrients is a key factor in driving the production of ligands and, therefore, the overall availability of dissolved iron. When the maximum biological production rate is perturbed by $\pm 50\%$ (Supplementary Information, Figs. S13, 14), there is no real change in atmospheric $CO_2$ compared to the standard simulations, indicating that the control run is producing emergent biological activity rates and concentration distributions based on nutrient limitation rather than artificially limited by parameter constraints. However, approaching MOC perturbations of double the control MOC (i.e., Southern Ocean upwelling strength around 40.0 Sv), atmospheric $CO_2$ anomalies are slightly sensitive to a decrease in the maximum biological production rate parameter, where ligand production and iron abundance are lower, and surface nutrients are not as completely consumed. The lifetime of ligands is also an important control on iron availability, macronutrient usage, and ocean inorganic carbon storage. With a 25% increase or decrease in degradation timescale ($1/\lambda = 283.0 \pm 71.0$ years) there are more significant changes in the resulting atmospheric $pCO_2$, up to 33 $\mu$atm more uptake and up to 48 $\mu$atm more outgassing, respectively (Supplementary Information, Figs. S15, 16). These lifetimes are consistent with the O(100) year residence time of partially-labile and semi-refractory dissolved organic carbon[41] and ligands in deep waters of the Atlantic[42,43]. Higher ligand lifetime increases the model parameter ratio $\gamma/\lambda$ from the data-optimized value of 4398.0 s to 5497.5 s, shifting the system towards an "iron-replete, macronutrient limited" regime[22], accumulating a larger pool of ligands due to slower rates of decay across all MOC strengths. Longer-lived ligands support a greater availability of iron, resulting in a more complete drawdown of surface macronutrients, particularly in the iron-limited Southern and Pacific Oceans. In contrast, decreasing ligand lifetime lowers the model $\gamma/\lambda$ ratio from the data-optimized value of 4398.0 s to 3298.5 s, shifting the system towards an "iron-limited, macronutrient replete" regime[22]. More rapid ligand decay reduces ligand standing stock and lowers the overall availability of iron, resulting in less complete macronutrient consumption in the Surface Southern and Pacific Oceans, again across all MOC strengths. Even though the gradient of the MOC-atmospheric $pCO_2$ relationship is sensitive to perturbations in the ligand lifetime, a negative overall correlation remains robust.

## Discussion

Biological activity in the ocean has a strong influence on ocean-atmosphere carbon partitioning and atmospheric $pCO_2$, which is intimately tied to rates of surface delivery and efficient consumption of macronutrients, such as nitrate and phosphate, as well as micronutrients, such as the trace metal iron, in the Southern Ocean. The biogeochemistry and bioavailability of micronutrients, in turn, may be influenced by the activity of microbes through the maintenance of a pool of organic chelating ligands that protect dissolved iron against loss by scavenging. Simulations of an idealized ocean biogeochemistry model demonstrate that the functional relationship between the MOC, Southern Ocean upwelling, and atmospheric $pCO_2$ can be fundamentally different depending on whether the organic ligand pool is fixed, or dynamically coupled to biological activity through the growth and decay of microbes. In an ocean with constant ligand concentration, efficient nutrient consumption, and atmospheric $CO_2$ drawdown is attained when the global MOC and Southern Ocean upwelling are slowed to reduce the delivery of inorganic carbon- and macronutrient-rich waters to the surface and limit the escape of $CO_2$ to the atmosphere. In contrast, in an ocean with dynamic ligand concentrations coupled to biological activity, a weak MOC results in decreased ligand production, scarce iron availability, and inefficient nutrient consumption, resulting in ocean carbon outgassing. A vigorous global

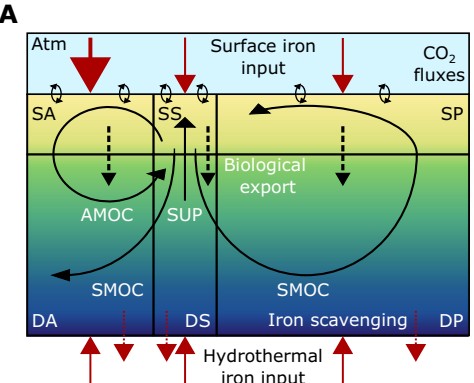
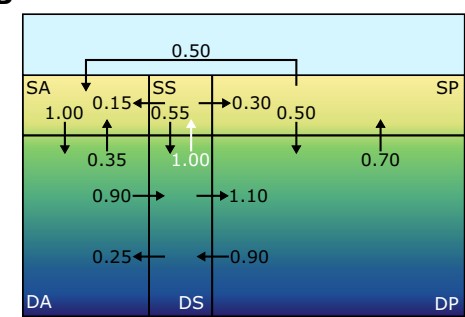

**Fig. 4 | Schematic of the "microCOSM" (micro-Carbon/Climate/Ocean Science Model) configuration.** The model is configured with six ocean boxes after[13,39] representing the Surface Atlantic (SA), Surface Southern (SS), Surface Pacific (SP), Deep Atlantic (DA), Deep Southern (DS), and Deep Pacific (DP) Oceans. **A** Two interconnected pathways of the largescale ocean circulation are represented (solid black arrows): Upwelling in the Southern Ocean (SUP) brings deep waters to the surface around Antarctica. These waters are then transported throughout the ocean in the Atlantic Meridional Overturning Circulation (AMOC), representing the path of North Atlantic Deep Waters, or the Southern Ocean Meridional Overturning Circulation (SMOC), representing the path of Antarctic Bottom Waters. Inorganic nutrients and carbon are transformed into organic matter in the surface, exported to depth, and remineralized (dashed black arrows). External sources of iron (solid red arrows) are added to the surface boxes (as spatially-varying dust or sedimentary input) or to the deep boxes (uniform hydrothermal vent input), while iron is lost by scavenging (dotted red arrows). Carbon dioxide is exchanged between the atmosphere and surface ocean (circular black arrows). **B** Transports between adjacent boxes are prescribed as a fraction of Southern Ocean upwelling (highlighted in white), which is varied over a wide range of values. AMOC transport is equivalent to Southern Ocean upwelling, although note that half of AMOC transport comes from the Surface Pacific, while SMOC is equivalent to 55% of Southern Ocean upwelling. Thus, the control model state with a Southern Ocean upwelling of 20.0 Sv has an AMOC of 20.0 Sv and a SMOC of 11.0 Sv. Fluxes were tuned to recreate the oceanic $^{14}C$ distribution[13]. Boxes are not drawn to scale in the vertical.

MOC with an enhanced supply of nutrients to the surface stimulates biological production and generates more ligands, enabling plentiful iron availability, resulting in efficient nutrient consumption as well as significantly reducing atmospheric $pCO_2$. This result is an independent mechanism from the glacial "iron hypothesis"[17], which invokes changes in external aeolian iron supply to the Southern Ocean in order to consume excess nutrients and prevent Southern Ocean $CO_2$ outgassing.

An increase in deep ocean inorganic carbon storage and more vigorous MOC could be compatible with the proxy record. Paleoceanographic evidence indicates more deep ocean biogenic carbon storage during the last glacial maximum [e.g., refs. 44–47], as well as more complete Southern Ocean nutrient utilization [e.g., refs. 15,46,48–51]. Still, the proxy records for ocean productivity[52] and past strength and configuration of the MOC are ambiguous[37,53]. For example, the change in the distribution of stable isotopes of carbon ($\delta^{13}C$) has been interpreted as a more sluggish MOC, with much less Southern Ocean upwelling during the last glacial maximum to trap isotopically-light carbon in the deep Southern Ocean [e.g., ref. 54]. However, the existence of well-defined gradients in the presence of more intense glacial tidal mixing [e.g., refs. 40] could instead suggest more vigorous deep ocean circulation[55,56]. Ocean models forced by paleoclimate boundary conditions have a broad range of responses for the AMOC[57–60], that should be interpreted with caution [e.g., ref. 61] but tend towards a stronger and slightly deeper glacial AMOC in latest results from phase 4 of the Paleoclimate Model Intercomparison Project [PMIP4,[30]]. Models that explicitly carry tracers representing multiple proxies also support varying conclusions about AMOC strength and the depth of the interface between the AMOC and SMOC[29,32,34–36,38], as well as ocean models that produce glacial state-estimates by quantitatively assimilating paleoceanographic observations[45,62–64]. According to results from the model with variable ligand concentrations, a slower glacial MOC [e.g., refs. 28,35,65,66] demands other processes to achieve lower atmospheric $CO_2$, such as iron fertilization[17] and extended Antarctic sea ice[67–69] to compensate for circulation-driven outgassing caused by low ligand and iron abundance.

Current changes in the MOC are also uncertain. Observational trends could indicate a slow down of the AMOC[70], but may be within the bounds of natural variability[71]. Changes in SMOC might also represent a trend or natural variability[72,73]. Future climate change projections do suggest a potential decline in the MOC between a half to one-third by 2100[74], which may be reinforced by poorly-represented Antarctic meltwater forcing[75,76]. The predicted decline in the global MOC already threatens the ocean's ameliorating effects on climate due to declining anthropogenic $CO_2$ uptake[77], but it could be further impeded if the MOC-ligand interplay detailed here is important, resulting in enhanced Southern Ocean natural carbon outgassing and accelerated atmospheric warming. The global Earth System Models used for these future climate projections do not routinely include the dynamic feedback between biological activity, ligand abundance, iron biogeochemistry, and the ocean carbon cycle explored here [e.g., refs. 78–83].

However, this is an idealized model, both in terms of simplified geometry and in capturing diverse processes that govern oceanic trace metal cycles, although the negative MOC-atmospheric $pCO_2$ correlation is robust to large changes in Southern Hemisphere surface iron input, reconfiguration of Atlantic circulation, mixing rates, maximum biological production rate, and prescribed ligand lifetime (Supplementary Information).

The treatment of iron cycling neglects colloidal and particulate phases of iron [e.g., ref. 84], parameterizes a generalized iron input rather than separating aeolian, sedimentary, riverine, glacial, and recycled sources [e.g., refs. 85–88], and carries a single dynamic pool of ligands without external sources rather than explicitly capturing varied pools like siderophores or terrestrially-derived humic substances that can also bind with iron[23,89,90]. Siderophores constitute a small fraction of ocean iron ligands [e.g., ref. 24] and may be produced by bacteria under iron stress to competitively capture iron, which could reduce the impact of the ligand-iron feedback. A similar framework of ligand cycling was extended with a second siderophore-type ligand in a box model[23], but the reinforcing feedback described here remained the dominant mechanism, rather than being largely damped as would be expected from siderophore-dominated ligand pool.

**Table 1 | microCOSM parameter values common to fixed and dynamic ligand configurations**

| Parameter | Symbol | Units | Common value(s) |
|---|---|---|---|
| No. of boxes | $n_b$ | | 6 |
| Box width | dx | km | $17.0 \times 10^6$, $17.0 \times 10^6$, $17.0 \times 10^6$, $17.0 \times 10^6$, $17.0 \times 10^6$, $17.0 \times 10^6$ |
| Box length | dy | km | $4.0 \times 10^6$, $4.0 \times 10^6$, $2.0 \times 10^6$, $2.0 \times 10^6$, $8.0 \times 10^6$, $8.0 \times 10^6$ |
| Box thickness | dz | m | 100.0 (2000.0), 3900.0 (2000.0), 100.0, 3900.0, 100.0, 3900.0 |
| MOC strength | $\psi$ | Sv | 20.0 (1.0–40.0) |
| AMOC fraction | $f_{AMOC}$ | | 1.00 |
| SMOC fraction | $f_{SMOC}$ | | 0.55 |
| Diffusive flux | $\kappa$ | Sv | 0.0 (3.0) |
| Max. productivity | $\alpha_{bio}$ | $\mu$mol P m$^{-3}$ s$^{-1}$ | 6.0 (3.0–9.0) |
| Stoichiometric ratios | $\mathcal{R}_{C:N:P:Fe}$ | mol mol$^{-1}$ | $106{:}16{:}1{:}1 \times 10^{-3}$ |
| P uptake coeff. | $k_P$ | $\mu$mol P kg$^{-1}$ | 0.1 |
| N uptake coeff. | $k_N$ | $\mu$mol N kg$^{-1}$ | 1.6 |
| Fe uptake coeff. | $k_{Fe}$ | nmol Fe kg$^{-1}$ | 0.1 |
| Light uptake coeff. | $k_I$ | W m$^{-2}$ | 30.0 |
| Iron input | $\mathcal{S}_{Fe}$ | g Fe m$^{-2}$ y$^{-1}$ | 0.300, 0.027, 0.003 ($1.50 \times 10^{-4}$–0.060), 0.013, 0.03, 0.054 |
| Solubility of iron | $s_{Fe}$ | | $2.5 \times 10^{-3}$ |
| Ligand stability coeff. | $\beta$ | (mol kg$^{-1}$)$^{-1}$ | $1.0 \times 10^9$ |
| Iron scavenging rate | $k_{scav}$ | s$^{-1}$ | $1.0 \times 10^{-7}$ |
| Average piston velocity | $k_{pis}$ | cm hr$^{-1}$ | 0.337 |
| Initial atmospheric CO$_2$ | $pCO_2^{atm}$ | $\mu$atm | 280.0 |
| Wind speed | $U$ | m s$^{-1}$ | 5.0, 0.0, 10.0, 0.0, 5.0, 0.0 |
| Open water fraction | $f_{open}$ | | 1.0, 0.0, 1.0, 0.0, 1.0, 0.0 |
| Seawater temperature | $\theta$ | °C | 20.0, 4.0, −1.0, −1.0, 20.0, 4.0 |
| Seawater salinity | S | g kg$^{-1}$ | 35.50, 35.50, 34.75, 34.75, 35.00, 35.00 |
| Initial carbon conc. | DIC | $\mu$mol C kg$^{-1}$ | 2100.0, 2400.0, 2100.0, 2400.0, 2100.0, 2400.0 |
| Initial alkalinity | ALK | $\mu$mol A kg$^{-1}$ | 2350.0, 2400.0, 2300.0, 2400.0, 2300.0, 2400.0 |
| Initial phosphate conc. | $PO_4^{3-}$ | $\mu$mol P kg$^{-1}$ | 0.0, 2.0, 2.0, 2.0, 0.0, 2.5 |
| Initial nitrate conc. | $NO_3^{1-}$ | $\mu$mol N kg$^{-1}$ | 0.0, 30.0, 30.0, 30.0, 0.0, 36.0 |
| Initial iron conc. | $Fe_T$ | nmol Fe kg$^{-1}$ | 0.0, 0.0, 0.0, 0.0, 0.0, 0.0 |

Lists indicate values in the SA, DA, SS, DS, SP, and DP boxes, respectively. Values in parentheses indicate perturbation experiment parameter ranges.

Uncertainty in the first-order impact of the MOC on the ocean-atmosphere carbon balance is a challenge to our understanding of the Earth system, particularly with regards to causes of glacial–interglacial CO$_2$ changes and climate cycles. This challenge still holds true even if the unexplored processes in this idealized model turn out to robustly decrease the slope of the dynamic ligand MOC-atmospheric pCO$_2$ relationship (Fig. 1b) to leave a weak correlation between the ocean's MOC and the atmosphere CO$_2$. Beyond external iron fertilization[17], these results suggest that dynamic, emergent feedbacks between ocean circulation, biological activity, and the marine iron cycle may be vital in shaping the ocean's response to climate and should be more wholly represented in the next generation of Earth System Models.

## Methods
### Model details and configuration
The micro-Carbon/Climate/Ocean Science Model ["microCOSM"][25] is an idealized box model of ocean circulation, inorganic carbon cycling, and trace metal biogeochemistry (Fig. 4).

The model is configured with a six-box geometry that captures the two interconnected cells of the global Meridional Overturning Circulation[4,13,39]. Transports between boxes are prescribed as a fraction of Southern Ocean upwelling, with fluxes in the control simulation prescribed to recreate the oceanic $^{14}C$ distribution[13]. Upwelling in the Southern Ocean brings deep waters to the surface around Antarctica. A fraction is subducted back into the deep Southern Ocean, and then transported laterally into the Deep Atlantic and Pacific basins before being partially upwelled and returned south, analogous to Antarctic Bottom Water formation in the Southern Ocean Meridional Overturning Circulation (SMOC) cell. The remaining fraction of Southern Ocean upwelling is transported laterally into the Surface Atlantic and Pacific boxes. In the Surface Atlantic box, Southern Ocean waters are joined by a flux from the Surface Pacific Ocean, representing flow east around the tip of South America and west around the tip of Southern Africa, and is subducted back into the deep ocean, analogous to North Atlantic Deep Water formation in the Atlantic Meridional Overturning Circulation (AMOC). Note that although AMOC strength is equal to Southern Ocean upwelling, half of the AMOC transport is derived from lateral fluxes originating in the Surface Pacific. An additional exchange between adjacent boxes due to mixing may also be prescribed.

Inorganic carbon, alkalinity, macronutrients (nitrate and phosphate), and the micronutrient iron are redistributed by this circulation and biological uptake in the surface, export of organic matter, and remineralization at depth. Biological activity is evaluated using Michaelis-Menten kinetics, with a maximum rate limited by macronutrient, iron, or light availability. Iron is added to the surface boxes, with significantly less iron added to the Surface Southern and Pacific Oceans than the Surface Atlantic Ocean to account for the remoteness and limited continental shelf area in the former and analogous to Saharan dust input or shelf sediment, and riverine sources in the latter region[9]. The deep ocean boxes receive an area-weighted source of iron equivalent to estimates of hydrothermal vent input[91].

**Table 2 | microCOSM parameter values specific for fixed and dynamic ligand model configurations**

| Parameter | Symbol | Units | Fixed Ligand | Dynamic Ligand[22] |
|---|---|---|---|---|
| Ligand feedback strength | $\gamma/\lambda$ | s | 0.0 | 4398.0 (3298.5–5497.5) |
| Ligand production fraction | $\gamma$ | mol L (mol P)$^{-1}$ | 0.0 | 0.0053 |
| Ligand lifetime (surface) | $1/\lambda$ | s | 0.0 | 87960000.0 (65970000.0–109950000.0) |
| Ligand degradation modifier | $\delta\lambda$ | | 0.0 | 1.0, 0.01, 1.0, 0.01, 1.0, 0.01 |
| Initial ligand conc. | $L_T$ | nmol L kg$^{-1}$ | 2.0 | 0.0, 0.0, 0.0, 0.0, 0.0, 0.0 |

Lists indicate values in the SA, DA, SS, DS, SP, and DP boxes, respectively. Values in parentheses indicate perturbation experiment parameter ranges.

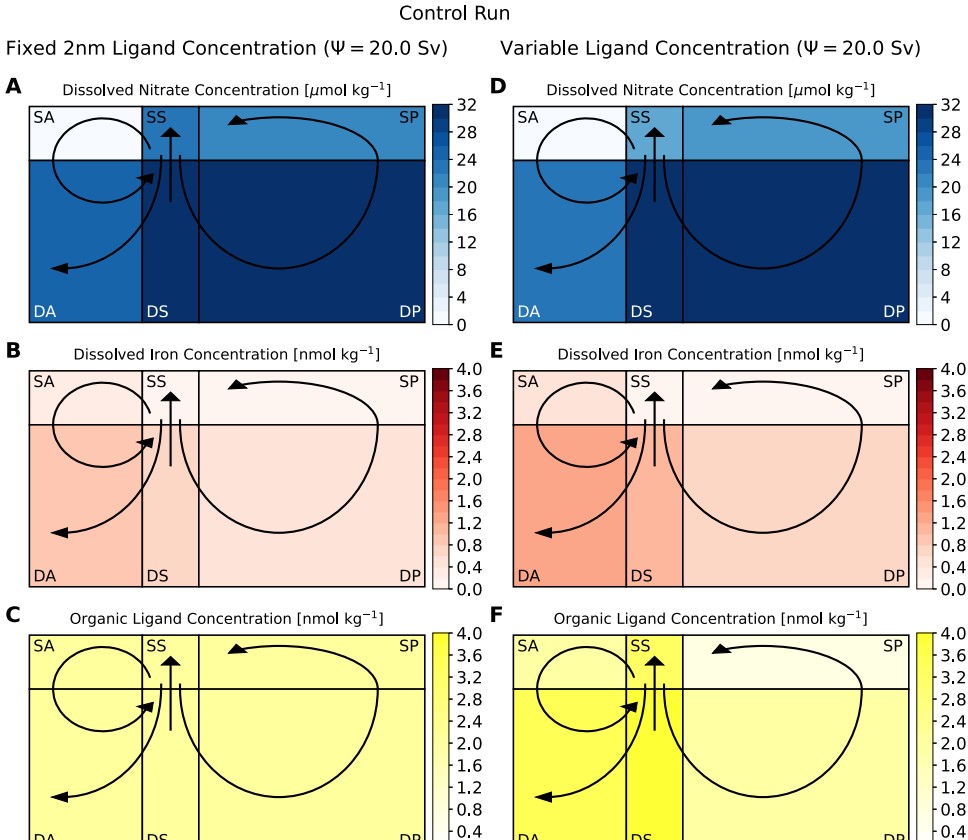

**Fig. 5 | Steady-state initial conditions for model simulations with the control Southern Ocean upwelling strength (20.0 Sv).** Ligand levels are represented as fixed, uniform concentrations (left column), or are allowed to dynamically varying (right column). Concentrations are (**A**, **D**) nitrate ($\mu$mol N kg$^{-1}$), (**B**, **E**) iron (nmol Fe kg$^{-1}$), and (**C**, **F**) organic ligands (nmol L kg$^{-1}$). Black arrows indicate the overturning circulation between boxes representing the Surface Atlantic (SA), Surface Southern (SS), Surface Pacific (SP), Deep Atlantic (DA), Deep Southern (DS), and Deep Pacific (DP) Oceans. Boxes are not drawn to scale in the vertical. Source data are provided as a Source Data file.

Iron undergoes complexation with a single pool of ligands that determines partitioning between free and protected iron[92]. Free iron is subject to scavenging and precipitation when above its low solubility[7]. Ligand concentration is either a constant and uniform concentration (2.0 nmol Fe kg$^{-1}$) in all six boxes or may represent a dynamic pool that is generated actively as a fraction of organic matter production near the surface, analogous to exudate release and siderophore production, as well passively due to deep ocean remineralization of sinking particulate organic matter, equivalent to the release of porphyrins and degraded protein remnants[22]. Ligand breakdown is modelled as a first-order process associated with depletion through photochemical degradation and heterotrophic microbial uptake in the upper ocean. Deep ocean ligand degradation is modified to be one hundred times slower due to lower microbial metabolism in cooler waters, and an order of magnitude decrease in bacterial abundance[93]. The ratio of ligand production to ligand loss parameters determines the system's

behavior, and this model uses the optimized values from a 10,000 member fit to observations from ref. 22. Ligand lifetime for this optimization is consistent with estimates of the residence time of ligands the deep Atlantic of $O(100)$ years[42,43].

Oceanic $pCO_2$ is solved at prescribed surface temperatures, and salinities[94], and air-sea fluxes of $CO_2$ are calculated using a prescribed gas transfer velocity. Carbon is exchanged between the ocean and an atmospheric box that can be set to a fixed $CO_2$ value (280.0 $\mu$atm) for model equilibration or allowed to vary in time depending on the air-sea carbon partitioning. Parameter values are given in Table 1.

Two reference experiments are spun up from arbitrary tracer starting concentrations, one with a fixed uniform ligand concentration of 2.0 nmol kg$^{-1}$ that is not subject to sources or sinks, and one with variable ligand concentration (initially 0.0 nmol kg$^{-1}$) subject to dynamic microbial feedbacks and redistribution by ocean transports. Parameter values for different ligand schemes are given in Table 2. The

model is integrated for 100 ky to a steady state with a Southern Ocean upwelling strength of 20.0 Sv (20.0 Sv AMOC and 11.0 Sv SMOC[5]). At steady state (Figs. 5, S1–4), the control simulations both capture the broad pattern of resource limitation in the real ocean: biological activity in the Surface Southern and Pacific Ocean boxes is limited by iron, with abundant macronutrient concentrations, while the Surface Atlantic box is instead limited by macronutrient supply, with excess iron availability. Ligand distributions in both cases are largely homogenous, reminiscent of the real ocean, including slightly elevated ligand abundance in the Southern Ocean upwelling box[27]. The parameter controlling the strength of the ocean's meridional overturning circulation ($\psi$, table 1) is varied across a broad range of overturning rates from an almost-collapsed MOC with 1.0 Sv Southern Ocean upwelling, through a weak MOC with 8.0 Sv of Southern Ocean upwelling to a vigorous double-strength circulation of 40.0 Sv. Each ensemble member is integrated to steady state for a further 100 ky[26].

## Evaluating completeness of nutrient utilization

The completeness, or efficiency, of nutrient consumption is evaluated focusing on the ratio of average surface concentration to the average concentration in the three deep ocean boxes. Incompletely or inefficiently utilized nutrients that are exported from the surface promote ocean $CO_2$ outgassing, whereas more completely or efficiently consumed nutrients prevent surface inorganic carbon escape by trapping biological carbon in the deep sea.

## Data availability

Model output and analysis workflow used in this study are available at (https://bit.ly/lauderdale-ligand-moc-pco2) and at the *Zenodo* open science archive[26]. The specific model version used in this analysis is included within this repository. Source data are provided with this paper.

## Code availability

Up-to-date *microCOSM* model code can be accessed via GitHub (https://bit.ly/microCOSM) and at the *Zenodo* open science archive[25].

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

## Acknowledgements

I am grateful to GEOTRACES program scientists, technicians, and crew for oceanic trace metal and ligand observations. The GEOTRACES 2021 Intermediate Data Product (IDP2021) represents an international collaboration endorsed by the Scientific Committee on Oceanic Research (SCOR). The many researchers and funding agencies responsible for data collection and quality control are thanked for their contributions to the IDP2021. I acknowledge funding from the National Science Foundation Partnerships for International Research and Education program (#1545859), the Simons Collaboration on Computational BIOgeochemical Modeling of marine EcosystemS, (CBIOMES #549931, awarded to M. Follows), NASA Carbon Cycle Sciences (#80NSSC22K0153), and the NSF GEO-NERC program (#2347991). I appreciate Mick Follows, Stephanie Dutkiewicz, Timothy Herbert, and Lauren Hinkel for discussions that greatly improved this manuscript. Thanks also to the reviewers for their thoughtful comments.

## Author contributions

J.M.L. conceived the study; J.M.L. developed the model; J.M.L. ran simulations and analyzed output; J.M.L. wrote the manuscript.

## Competing interests

The authors declare no competing interests.
