## [Peer Review File · Nature Communications]

REVIEWER COMMENTS

Reviewer #1 (Remarks to the Author):

The author of this manuscript creates a thought experiment, consisting of four boxes that mimic physical and biogeochemical exchange between the surface Antarctic, sub-Antarctic, rest of the surface Atlantic, and Deep Atlantic regions. Their focus is the effects of changes in AMOC transport and iron dynamics on atmospheric CO₂. Their main finding is that variable ligand concentrations apparently produce an ocean response of decrease in atmospheric CO₂ with enhanced AMOC. I think this work is not ready for publication in Nature Communications, due to the following issues:

- 1) A box model that pretends to find a response of atmospheric CO₂ to ocean dynamic changes, particularly Southern Ocean upwelling, should include a box associated with the Indo-Pacific basin, which holds the most carbon content of the ocean. That box could be connected to the Antarctic and Southern Ocean boxes as per Fig. 4.
- 2) The author employs most of the discussion to relate their findings with the box model experiment to the ocean configuration of the LGM. However, they fail to realize that the main difference between the Atlantic deep water mass configuration in the Holocene and in the LGM is not AMOC transport, but the depth of the interface between what they call AMOC and SMOC (see, e.g., Gebbie, 2014 *Paleoceanography and Paleoclimatology*; Gu et al., 2020 *Earth and Planetary Science Letters*; Muglia and Schmittner, 2021, *Quaternary Science Reviews*). In this context, if the author wants to draw a line between the results of their box model experiments and what we know from proxies for the LGM deep Atlantic water mass configuration, my advice would be to follow a different experiment approach: To check the effects of varying the relative size of "Deep" and "Atlantic" boxes of their box model. Otherwise, my advice would be to be much more cautious on linking their results with LGM deep ocean configuration as we know it from proxies.
- 3) A comment on Figure 1B, that may be a misunderstanding by me, but nevertheless I wish to clarify: Based on the range given by different past publications, a realistic AMOC transport for the LGM should sit in the range between 8 and 25 Sv. It is true that in the box model with default atmospheric iron flux a surprising CO₂-AMOC relationship arises, such that with enhanced SO upwelling we have less atmospheric CO₂. However, if I understand what the blue shading means, if we increase the atmospheric Fe flux the relationship becomes like in Fig. 1A for AMOC transports lower than 20 Sv. In this context, if the LGM had higher dust deposition over the Southern Ocean, the real system was probably more similar to an "increased surface iron input" scenario in the box model, and according to the box model results of Fig. 1B, the response would be lower atmospheric CO₂ with weaker AMOC.

About the plots:

Fig 1: For clarity, express the direction of iron input and biological production changes of each side of the blue and orange envelopes.

Figures 2 and 3 are useful, but I think that the analysis between lines 94 and 207 would benefit from scatter plots more similar to Figure 1. The authors could plot dissolved nitrate, dissolved iron, and ligand concentrations for ea

For original research articles submitted on or after 1st November 2022 that are accepted for publication, the reviewer comments to authors and the author rebuttal letters of revised versions will be published online as a supplementary "peer review file". For original research articles submitted before 1st November 2022, the authors can choose to opt-in or opt-out of the transparent peer-review scheme and have the option to publish the reviewer comments to the authors as well as the author rebuttal letters along with the paper if this is accepted for publication.

If you choose to be acknowledged by name as outlined above, we wch box as a function of Southern Ocean upwelling strength, in three separate plots (one plot for each tracer). These plots could go in the supplementary materials.

Minor comments:

Lines 32-38: The Indo-Pacific Ocean is omitted in this discussion, and should be included.

Line 38: Rather than "approximately equal", I would say "same order of magnitude", or "similar magnitude".

Line 235: The stronger AMOC simulated by PMIP3 models that is used in reference [39] should not be used as evidence of a stronger AMOC during the LGM, as it is stated in that paper and in Marzocchi and Jansen (2017), *Geophysical Research Letters*, that those models present strong AMOC partly as a result of not having acquired an equilibrium state.

Line 236: While it is true that references 41-43 point towards and enhanced AMOC during the LGM, there exist other works equally well constrained by proxies that show a weak-transport LGM AMOC (Menviel et al., 2017, *Paleoceanography and Paleoclimatology*, Muglia et al., 2018, *Earth and Planetary Science Letters*). In fact, both Gu et al., 2020, *Earth and Planetary Science Letters* and Muglia and Schmittner, 2021, *Quaternary Science Reviews*, show that current proxies are not able to constrain AMOC strength in models, but only AMOC geometry (i.e., its depth).

Reviewer #2 (Remarks to the Author):

see attached

Ocean iron cycle feedbacks decouple atmospheric CO₂ from meridional overturning circulation changes

General comments: The manuscript entitled, “Ocean iron feedbacks decouple atmospheric CO₂ from meridional overturning circulation changes” written by Jonathan M. Lauderdale, discusses how including a variable organic iron-binding ligand pool in a simple climate model impacts the atmospheric CO₂ changes associated with changes in meridional overturning circulation (MOC). In the model, if MOC is slowed and you have a fixed ligand pool, you get less CO₂ outgassing from the ocean (lower atmospheric CO₂) and this trend is reversed if you have a variable organic ligand pool in the model. This suggests that simply changing the organic iron-binding ligands can reverse the typical trends associated with glacial-interglacial changes in MOC. This pattern was robust even with a changing iron supply, suggesting that ligands might have a larger effect on CO₂ outgassing from the ocean in the past than simple changes in iron supply such as from dust, sediments or ice.

This is a really interesting result, and builds on previous work by this author and some other recent work focused on changes in iron recycling over paleo timescales (Rafter et al. 2017). Overall, I really enjoyed reading the paper and thought it was well-written and well-articulated to a broad audience.

The only major thing I think the paper is missing, is a bit more discussion on the residence time of the ligands used in the model. I know this is discussed in Lauderdale et al. (2020), but it is a very important point for this work. For example, although there is very little data on the residence time of ligands that has been published in the literature, there are a few estimates that span a large range. Surface iron-ligand complexes (iron in siderophores) have turnover times on the order of 8 -300 days (Bundy et al. 2022) and deep ocean ligands have turnover times on the order of 779-1039 years (Gerringa et al. 2015). So for example, in the model if an increase in biological production occurs and more ligands are produced (variable ligand model), if these ligands have a shorter turnover time that is similar to observations, does ligand production still allow for more iron to accumulate and ultimately upwelled in the Southern Ocean? I guess I am wondering if surface ligand production of short-lived ligands would ultimately have the same effect in the model, since some amount of the ligands produced in the surface waters in the model appear to make it to the deep ocean (similar to the deep North Atlantic) and are partially responsible for allowing hydrothermal iron to resist scavenging, and accumulate. If I am understanding Lauderdale et al. (2020) correctly, then I think this is partially already considered in this model, in that ligands in the surface have shorter residence times but are degraded into ligands in the deeper waters that then have longer residence times. If this interpretation is correct, then I think that is probably the best we can do in a model of ligands at this stage. I do however, think a more thorough discussion of this should be included in the main text because I suspect it is key to the results.

Overall I think this is a very thought-provoking paper and I really enjoyed it. I have a few additional very minor comments below.

Specific comments:

Line 31: Might want to specify here that MOC returns nutrients to surface waters on very long timescales in order to distinguish it from Ekman pumping.

Line 39: Might want to specify “inorganic carbon” here and in some other places throughout the manuscript, where you mean dissolved inorganic carbon but have just written carbon.

Lines 87-92: It is hard for me to conceptualize changes in $p\text{CO}_2$ here with Sverdrups. Can you give some examples of total atmospheric $p\text{CO}_2$ change with a realistic change in MOC strength for example? Might help to put it in perspective if presented based on total atmospheric $p\text{CO}_2$ for the average reader.

Lines 101-102: In the weaker MOC scenario, is there a way to separate the fact that lower dissolved organic carbon concentrations would be upwelled to the surface (even in the absence of biology) and the lowering of surface DIC due to more complete nutrient consumption? I am wondering if lower DIC in the weaker MOC scenario is mostly an abiotic processes, since less DIC would be upwelled to the surface, thus causing less outgassing to the atmosphere.

Line 133: What do you mean here by iron feedback?

Line 149: Capitalize “substantially”

References

Bundy, R.M., Manck, L.E., Boiteau, R.M., Park, J., Delong, E.F., Hawco, N.J., Church, M.J., Saito, M. and Repeta, D.J., 2022. Seasonal siderophore uptake and biosynthesis associated with carbon flux at Station ALOHA. *bioRxiv*, pp.2022-10.

Gerringa, L.J., Rijkenberg, M.J., Schoemann, V., Laan, P. and De Baar, H.J., 2015. Organic complexation of iron in the West Atlantic Ocean. *Marine Chemistry*, 177, pp.434-446.

Lauderdale, J.M., Braakman, R., Forget, G., Dutkiewicz, S. and Follows, M.J., 2020. Microbial feedbacks optimize ocean iron availability. *Proceedings of the National Academy of Sciences*, 117(9), pp.4842-4849.

Rafter, P.A., Sigman, D.M. and Mackey, K.R., 2017. Recycled iron fuels new production in the eastern equatorial Pacific Ocean. *Nature communications*, 8(1), p.1100.

Reviewer #1 comments:

The author of this manuscript creates a thought experiment, consisting of four boxes that mimic physical and biogeochemical exchange between the surface Antarctic, sub-Antarctic, rest of the surface Atlantic, and Deep Atlantic regions. Their focus is the effects of changes in AMOC transport and iron dynamics on atmospheric CO₂. Their main finding is that variable ligand concentrations apparently produce an ocean response of decrease in atmospheric CO₂ with enhanced AMOC. I think this work is not ready for publication in Nature Communications, due to the following issues:

I appreciate reviewer 1's thoughtful comments.

1) A box model that pretends to find a response of atmospheric CO₂ to ocean dynamic changes, particularly Southern Ocean upwelling, should include a box associated with the Indo-Pacific basin, which holds the most carbon content of the ocean. That box could be connected to the Antarctic and Southern Ocean boxes as per Fig. 4.

Although I believe there is still an important role for single-basin models in understanding the link between marine biogeochemistry, the ocean carbon cycle, and atmospheric CO₂ (from [1] to [2]), I have reconfigured the box model to include a Pacific basin by adding an additional two boxes following [3–6]. Fluxes between boxes in this new six-box model are constrained by [3] to recreate the observed ¹⁴C distribution in the real ocean. The control experiment has a Southern Ocean upwelling strength of 20 Sv, with a net Atlantic Meridional Overturning Circulation (AMOC), driven by sinking of dense waters in the North Atlantic of 20 Sv, and an 11 Sv net Southern Ocean Meridional Overturning Circulation (SMOC), comprising dense bottom water sinking adjacent to Antarctica. I note that although AMOC strength is equal to Southern Ocean upwelling, half of the AMOC transport is derived from lateral fluxes originating in the Surface Pacific.

Varying Southern Ocean upwelling over a range of 8–40 Sv reproduces the same fundamentally different relationships between ocean circulation strength and atmospheric CO₂ levels with fixed and dynamically varying ligand concentrations as in the original manuscript. Figures and results throughout the manuscript have been updated accordingly.

2) The author employs most of the discussion to relate their findings with the box model experiment to the ocean configuration of the LGM. However, they fail to realize that the main difference between the Atlantic deep water mass configuration in the Holocene and in the LGM is not AMOC transport, but the depth of the interface between what they call AMOC and SMOC (see, e.g., Gebbie,

2014 Paleooceanography and Paleoclimatology; Gu et al., 2020 Earth and Planetary Science Letters; Muglia and Schmittner, 2021, Quaternary Science Reviews). In this context, if the author wants to draw a line between the results of their box model experiments and what we know from proxies for the LGM deep Atlantic water mass configuration, my advice would be to follow a different experiment approach: To check the effects of varying the relative size of "Deep" and "Atlantic" boxes of their box model. Otherwise, my advice would be to be much more cautious on linking their results with LGM deep ocean configuration as we know it from proxies.

I recognize that highly idealized models have limitations. The new six-box model has a Surface Atlantic depth of 100 m, which conceptually represents the euphotic layer, but since northward AMOC flow occurs in the surface box and southward AMOC flow occurs in the deep box, the depth of this interface could be interpreted as a shallow depth of the AMOC maximum, rather than depth extent of the AMOC.

I have added a sensitivity experiment in which the interface between upper ocean and deep ocean in the Atlantic is increased to 2000 m (out of a total depth of 4000 m). I have included a summary of these simulations in the "Robustness" section (lines 196–204), with the details in the Supplementary Information (text and Figs. S5–8). These experiments indicate that this perturbation affects the strong MOC experiments much more than those with a weak MOC: there is reduced CO₂ outgassing for fixed ligand simulations (due to changes in Surface and Deep Atlantic residence times that increase accumulation of iron) and lower CO₂ uptake for dynamic ligand simulations (associated with changes in Surface and Deep Atlantic residence times that increase ligand degradation and reduce overall iron concentrations). These additional experiments have now covered weak-shallow, strong-shallow, weak-deep, and strong-deep scenarios, but across the ensemble, the opposing signs of MOC–atmospheric pCO₂ relationships between fixed and dynamic ligand concentrations remains robust.

I have also revised the discussion to be more cautious in the interpretation of my results in this context, and to better express the intent of this section. I have made the topic sentence of this paragraph clearer (lines 257–258), stating: *"An increase in deep ocean carbon storage and more vigorous MOC could be compatible with the proxy record"*. I end this section with (lines 272–274): *"a slower glacial MOC [e.g., 7–10] demands other processes to achieve lower atmospheric CO₂, such as iron fertilization [11] and extended Antarctic sea ice [2, 12, 13] to compensate for circulation-driven outgassing caused by low ligand and iron abundance."* I believe this strikes a balance across different groups producing diverse results regarding reconstructions of the LGM AMOC.

3) A comment on Figure 1B, that may be a misunderstanding by me, but nevertheless I wish to clarify: Based on the range given by different past publications, a realistic AMOC transport for the LGM should sit in the range between 8 and 25 Sv. It is true that in the box model with default atmospheric iron flux a surprising CO₂-AMOC relationship arises, such that with enhanced SO upwelling we have less atmospheric CO₂. However, if I understand what the blue shading means, if we increase the atmospheric Fe flux the relationship becomes like in Fig. 1A for AMOC transports lower than 20 Sv. In this context, if the LGM had higher dust deposition over the Southern Ocean, the real system was probably more similar to an "increased surface iron input" scenario in the box model, and according to the box model results of Fig. 1B, the response would be lower atmospheric CO₂ with weaker AMOC.

In the previous model, the AMOC strength was 50% of Southern Ocean upwelling, so the realistic range of LGM AMOC strengths of 8–25 Sv would have sat to the right of the blue "enhanced iron deposition" shaded region (i.e. 16–50 Sv in old Figure 1), supporting the negative Southern Ocean upwelling vs atmospheric CO₂ relationship for dynamic ligands.

I have highlighted the realistic range of LGM AMOC strengths of 8–25 Sv [e.g. 8, 14–17] pointed out by the reviewer in my figures to aid interpretation. In the new model, I have expanded the reasoning behind not focussing on the very lowest rates of Southern Ocean upwelling between 1–8 Sv (lines 90–92): *"At 8.0 Sv, the vast deep Pacific Ocean is ventilated by a net transport of only 1.6 Sv, creating a residence time on the order of 10 kyrs that effectively traps carbon and nutrients, leading to decline in atmospheric pCO₂ of ~50–60 μatm, irrespective of surface macro- or micronutrient limitation."*

About the plots:

Fig 1: For clarity, express the direction of iron input and biological production changes of each side of the blue and orange envelopes.

For Figure 1 and the sensitivity experiment figures in the Supplementary Information, I have indicated the direction of iron input and other perturbation changes on their respective figures using different color lines and symbols (+ or –).

Figures 2 and 3 are useful, but I think that the analysis between lines 94 and 207 would benefit from scatter plots more similar to Figure 1. The authors could plot dissolved nitrate, dissolved iron,

and ligand concentrations for each box as a function of Southern Ocean upwelling strength, in three separate plots (one plot for each tracer). These plots could go in the supplementary materials.

I have included scatter plots of atmospheric $p\text{CO}_2$, export production, and tracer concentrations as Supplementary Information. I did not include these plots in the main manuscript for clarity, presentation, and readability reasons, as they are very busy (six boxes, three tracers, four panels for export production and one panel for atmospheric $p\text{CO}_2$, for a total of 23 panels), and there are twice as many panels again for fixed and dynamic ligands.

Minor comments:

Lines 32-38: The Indo-Pacific Ocean is omitted in this discussion, and should be included.

I have added a comment about the Pacific Ocean (lines 36–38).

Line 38: Rather than "approximately equal", I would say "same order of magnitude", or "similar magnitude".

I have revised this phrasing (line 40).

Line 235: The stronger AMOC simulated by PMIP3 models that is used in reference [39] should not be used as evidence of a stronger AMOC during the LGM, as it is stated in that paper and in Marzocchi and Jansen (2017), *Geophysical Research Letters*, that those models present strong AMOC partly as a result of not having acquired an equilibrium state.

Line 236: While it is true that references 41-43 point towards and enhanced AMOC during the LGM, there exist other works equally well constrained by proxies that show a weak-transport LGM AMOC (Menviel et al., 2017, *Paleoceanography and Paleoclimatology*, Muglia et al., 2018, *Earth and Planetary Science Letters*). In fact, both Gu et al., 2020, *Earth and Planetary Science Letters* and Muglia and Schmittner, 2021, *Quaternary Science Reviews*, show that current proxies are not able to constrain AMOC strength in models, but only AMOC geometry (i.e., its depth).

I appreciate the reviewer for their literature suggestions and I have included these more recent interpretations, as well as reworded this section of my discussion. See comment #3 above.

Reviewer #2 comments:

General comments: The manuscript entitled, "Ocean iron feedbacks decouple atmospheric CO_2 from meridional overturning circulation changes" written by Jonathan M. Lauderdale, discusses how

including a variable organic iron-binding ligand pool in a simple climate model impacts the atmospheric CO₂ changes associated with changes in meridional overturning circulation (MOC). In the model, if MOC is slowed and you have a fixed ligand pool, you get less CO₂ outgassing from the ocean (lower atmospheric CO₂) and this trend is reversed if you have a variable organic ligand pool in the model. This suggests that simply changing the organic iron-binding ligands can reverse the typical trends associated with glacial-interglacial changes in MOC. This pattern was robust even with a changing iron supply, suggesting that ligands might have a larger effect on CO₂ outgassing from the ocean in the past than simple changes in iron supply such as from dust, sediments or ice. This is a really interesting result, and builds on previous work by this author and some other recent work focused on changes in iron recycling over paleo timescales (Rafter et al. 2017). Overall, I really enjoyed reading the paper and thought it was well-written and well-articulated to a broad audience.

I am grateful to reviewer 2 for their encouraging remarks.

The only major thing I think the paper is missing, is a bit more discussion on the residence time of the ligands used in the model. I know this is discussed in Lauderdale et al. (2020), but it is a very important point for this work. For example, although there is very little data on the residence time of ligands that has been published in the literature, there are a few estimates that span a large range. Surface iron-ligand complexes (iron in siderophores) have turnover times on the order of 8 -300 days (Bundy et al. 2022) and deep ocean ligands have turnover times on the order of 779-1039 years (Gerringa et al. 2015). So for example, in the model if an increase in biological production occurs and more ligands are produced (variable ligand model), if these ligands have a shorter turnover time that is similar to observations, does ligand production still allow for more iron to accumulate and ultimately upwelled in the Southern Ocean? I guess I am wondering if surface ligand production of short-lived ligands would ultimately have the same effect in the model, since some amount of the ligands produced in the surface waters in the model appear to make it to the deep ocean (similar to the deep North Atlantic) and are partially responsible for allowing hydrothermal iron to resist scavenging, and accumulate. If I am understanding Lauderdale et al. (2020) correctly, then I think this is partially already considered in this model, in that ligands in the surface have shorter residence times but are degraded into ligands in the deeper waters that then have longer residence times. If this interpretation is correct, then I think that is probably the best we can do in a model of ligands

at this stage. I do however, think a more thorough discussion of this should be included in the main text because I suspect it is key to the results.

I have included a new sensitivity experiment examining an increase or decrease in the ligand lifetime (lines 221–236). As the reviewer points out, ligand lifetime is a key factor in determining the *magnitude* of atmospheric pCO₂ change for the dynamic ligand experiments (Supplementary text and figs. S15–16): across the range of MOC values, higher ligand lifetime shifts the system towards an “iron-replete, macronutrient limited” regime (e.g. Fig.4 in [18]), accumulating a larger pool of ligands due to slower rates of decay. Longer-lived ligands support a greater availability of iron, resulting in more-complete drawdown of surface macronutrients in the iron-limited regions and generally lower atmospheric pCO₂. With reduced ligand lifetime, the system shifts towards an “iron-limited, macronutrient replete” regime (e.g. Fig.4 in [18]), with a generally lower ligand concentration due to faster rates of decay. This results in lower availability of iron, enhanced iron limitation, and therefore less-complete consumption of macronutrients and higher atmospheric pCO₂. The *sign* of the change in the MOC–pCO₂ relationship remains negative, that is, dynamic ligand concentrations still reverse the relationship between overturning circulation and atmospheric pCO₂ compared to uniform ligand concentrations.

Overall I think this is a very thought-provoking paper and I really enjoyed it. I have a few additional very minor comments below. Specific comments:

Line 31: Might want to specify here that MOC returns nutrients to surface waters on very long timescales in order to distinguish it from Ekman pumping.

I have clarified the timescales involved (line 31).

Line 39: Might want to specify “inorganic carbon” here and in some other places throughout the manuscript, where you mean dissolved inorganic carbon but have just written carbon.

I have made this distinction throughout the manuscript.

Lines 87-92: It is hard for me to conceptualize changes in pCO₂ here with Sverdrups. Can you give some examples of total atmospheric pCO₂ change with a realistic change in MOC strength for example? Might help to put it in perspective if presented based on total atmospheric pCO₂ for the average reader.

Agreed, although it is difficult to determine what a “realistic change in MOC strength” might be. However, as Reviewer 1 points out, the range of potential LGM AMOC strengths is 8–25 Sv [e.g. 8, 14–17], thus I have included a bulk estimate of atmospheric pCO₂ change based on these figures (lines 85–87 and lines 96–97).

Lines 101-102: In the weaker MOC scenario, is there a way to separate the fact that lower dissolved organic carbon concentrations would be upwelled to the surface (even in the absence of biology) and the lowering of surface DIC due to more complete nutrient consumption? I am wondering if lower DIC in the weaker MOC scenario is mostly an abiotic processes, since less DIC would be upwelled to the surface, thus causing less outgassing to the atmosphere.

Organic carbon could play a modest role in atmospheric $p\text{CO}_2$ changes in its role as a carbon sink, in addition to its role here as a pool of chelating ligands, which are present at 3 orders of magnitude lower concentrations than DIC (nmol L^{-1} compared to $\mu\text{mol C kg}^{-1}$). This model does not include a separate organic carbon reservoir, partly due to the coarseness of vertical model resolution since all biological production is exported to the deep ocean and fully remineralized there, and the relatively long timescales involved in the MOC.

Line 133: What do you mean here by iron feedback?

I have removed the phrase “iron feedback” from this sentence as it was confusing.

Line 149: Capitalize “substantially”

I have made this modification to the manuscript

References

- [1] Sarmiento, J. L. & Toggweiler, J. R. A new model for the role of the oceans in determining atmospheric pCO₂. *Nature* **308**, 621–624 (1984).
- [2] Marzocchi, A. & Jansen, M. F. Global cooling linked to increased glacial carbon storage via changes in Antarctic sea ice. *Nat. Geosci.* **12**, 1001–1005 (2019). URL <https://www.nature.com/articles/s41561-019-0466-8>.
- [3] Broecker, W. S. & Peng, T.-H. Carbon cycle: 1985 glacial to interglacial changes in the operation of the global carbon cycle. *Radiocarbon* **28**, 309–327 (1986). URL <https://www.cambridge.org/core/product/453E35DB3657CEF83F8B9E247287FF38>.
- [4] Broecker, W. S. & Peng, T.-H. The role of CaCO₃ compensation in the glacial to interglacial atmospheric CO₂ change. *Global Biogeochemical Cycles* **1**, 15–29 (1987). URL <https://doi.org/10.1029/GB001i001p00015>.
- [5] Broecker, W. S. & Peng, T.-H. The cause of the glacial to interglacial atmospheric CO₂ change: A polar alkalinity hypothesis. *Global Biogeochem. Cycles* **3**, 215–239 (1989). URL <https://doi.org/10.1029/GB003i003p00215>.
- [6] Parekh, P., Follows, M. J. & Boyle, E. Modeling the global ocean iron cycle. *Global Biogeochem. Cycles* **18**, GB1002 (2004). URL <http://doi.org/10.1029/2003GB002061>.
- [7] Meniel, L. *et al.* Poorly ventilated deep ocean at the Last Glacial Maximum inferred from carbon isotopes: A data-model comparison study. *Paleoceanography* **32**, 2–17 (2017). URL <https://onlinelibrary.wiley.com/doi/abs/10.1002/2016PA003024>.
- [8] Muglia, J., Skinner, L. C. & Schmittner, A. Weak overturning circulation and high Southern Ocean nutrient utilization maximized glacial ocean carbon. *Earth and Planetary Science Letters* **496**, 47–56 (2018). URL <https://www.sciencedirect.com/science/article/pii/S0012821X18303212>.
- [9] Meniel, L. C. *et al.* Enhanced Mid-depth Southward Transport in the Northeast Atlantic at the Last Glacial Maximum Despite a Weaker AMOC. *Paleoceanogr. Paleoclimatology* **35**, e2019PA003793 (2020). URL <https://onlinelibrary.wiley.com/doi/abs/10.1029/2019PA003793>.

- [10] Rafter, P. A. *et al.* Global reorganization of deep-sea circulation and carbon storage after the last ice age. *Science Advances* **8**, eabq5434 (2022). URL <https://www.science.org/doi/abs/10.1126/sciadv.abq5434>.
- [11] Martin, J. H. Glacial–interglacial CO₂ change: the iron hypothesis. *Paleoceanography* **5**, 1–13 (1990).
- [12] Stephens, B. B. & Keeling, R. F. The influence of Antarctic sea ice on glacial-interglacial CO₂ variations. *Nature* **404**, 171–174 (2000).
- [13] Ferrari, R. *et al.* Antarctic sea ice control on ocean circulation in present and glacial climates. *Proc. Nat. Acad. Sci.* **111**, 8753–8758 (2014). URL <http://www.pnas.org/content/111/24/8753.abstract>.
- [14] Muglia, J. & Schmittner, A. Carbon isotope constraints on glacial Atlantic meridional overturning: Strength vs depth. *Quaternary Science Reviews* **257**, 106844 (2021). URL <https://www.sciencedirect.com/science/article/pii/S0277379121000512>.
- [15] Kageyama, M. *et al.* The PMIP4 Last Glacial Maximum experiments: Preliminary results and comparison with the PMIP3 simulations. *Clim. Past* **17**, 1065–1089 (2021). URL <https://cp.copernicus.org/articles/17/1065/2021/>.
- [16] Oka, A. *et al.* Glacial mode shift of the Atlantic meridional overturning circulation by warming over the Southern Ocean. *Commun. Earth Environ.* **2**, 1–8 (2021). URL <https://www.nature.com/articles/s43247-021-00226-3>.
- [17] Pöppelmeier, F., Jeltsch-Thömmes, A., Lippold, J., Joos, F. & Stocker, T. F. Multi-proxy constraints on Atlantic circulation dynamics since the last ice age. *Nat. Geosci.* **16**, 349–356 (2023). URL <https://www.nature.com/articles/s41561-023-01140-3>.
- [18] Lauderdale, J. M., Braakman, R., Forget, G., Dutkiewicz, S. & Follows, M. J. Microbial feedbacks optimize ocean iron availability. *Proc. Nat. Acad. Sci.* **117**, 4842 (2020). URL <http://www.pnas.org/content/117/9/4842.abstract>.

REVIEWERS' COMMENTS

Reviewer #1 (Remarks to the Author):

The author has satisfactorily addressed all my comments. The supplementary materials have the references broken. He should correct that.

Reviewer #2 (Remarks to the Author):

I thank the author for their thoughtful responses to the reviewer comments and find the manuscript has greatly improved.

Reviewer #1 comments:

The author has satisfactorily addressed all my comments. The supplementary materials have the references broken. He should correct that.

Thank you to Reviewer #1 for their time reviewing my manuscript, and their insightful comments. I have addressed the Supplementary Information links.

Reviewer #2 comments:

I thank the author for their thoughtful responses to the reviewer comments and find the manuscript has greatly improved.

Thank you to Reviewer #2, I appreciate their constructive comments and feedback that helped improve my manuscript.